# The Paradoxical Effects of Serum Amyloid-P Component on Disseminated Candidiasis

**DOI:** 10.3390/pathogens11111304

**Published:** 2022-11-06

**Authors:** Stephen A. Klotz, Peter N. Lipke

**Affiliations:** 1Division of Infectious Diseases, Department of Medicine, University of Arizona, Tucson, AZ 85724, USA; 2Department of Biology, Brooklyn College, CUNY, New York, NY 11210, USA

**Keywords:** serum amyloid-P component (SAP), C-reactive protein (CRP), miridesap, disseminated candidiasis, murine model of disseminated candidiasis, invasive fungi, functional amyloid, pattern recognition receptor

## Abstract

Serum amyloid P component (SAP) may play an important role in human fungal diseases. SAP binds to functional amyloid on the fungal surface and masks fungi from host immune processes, skewing the macrophage population from the pro-inflammatory M1 to the quiescent M2 type. We assessed the role of SAP in a murine model of disseminated candidiasis. Mice were injected with human SAP subcutaneously (SQ) followed by intravenous injection of *Candida albicans*. Male, BALBcJ mice were administered 2 mg human SAP or the homologous human pro-inflammatory pentraxin CRP, SQ on day −1 followed by 1 mg on days 0 thru 4; yeast cells were administered intravenously on day 0. Mice not receiving a pentraxin were morbid on day 1, surviving 4–7 days. Mice administered SAP survived longer than mice receiving yeast cells alone (*p* < 0.022), although all mice died. Mice given CRP died faster than mice receiving yeast cells alone (*p* < 0.017). Miridesap is a molecule that avidly binds SAP, following which the complex is broken down by the liver. Miridesap administered in the drinking water removed SAP from the serum and yeast cells and significantly prolonged the life of mice (*p* < 0.020). Some were “cured” of candidiasis. SAP administered early in the septic process provided short-lived benefit to mice, probably by blunting cytokine secretion associated with disseminated candidiasis. The most important finding was that removal of SAP with miridesap led to prolonged survival by removing SAP and preventing its dampening effects on the host immune response.

## 1. Introduction

Serum amyloid P component (SAP) is a phylogenetically conserved serum protein made in the liver. SAP is considered by some authorities to be a Pattern Recognition Receptor within the innate immune system [1] and is present constitutively in human serum. In humans it circulates at constant median concentrations of 32 µg/mL in men and 21 µg/mL in women [2]. Its known ligands include amyloid fibrils of any composition, and the fibrils are protected from proteolysis by SAP [3]. SAP is a powerful anti-opsonin in systemic bacterial infections in mice [4], and this is true with *C. albicans* as well [5]. Although many claims have been made about the function of SAP, its true purpose still eludes us [2]. SAP is a normal component of the interstitial matrix but its high prevalence in amyloid deposits, whether intra- or extracellular, is a remarkable feature. One of the hallmarks of amyloid histology is the absence of a host response to the abnormal proteins which are bound by SAP. In mice, SAP functions similarly, but it is an acute-phase reactant, and the serum concentration increases many-fold following infection, inflammation, or tumor growth [6].

Clinical trials involving administration or elimination of SAP provide evidence that SAP may be beneficial in some circumstances but is deleterious in others. For example, in idiopathic pulmonary fibrosis, long-term intravenous treatment with pharmacologic doses of recombinant human SAP stabilizes pulmonary function in patients [7]. However, SAP is best known for its deleterious properties, especially in its role in the amyloidoses. SAP is a universal constituent of amyloid deposits and protects amyloid fibers from proteolysis and subsequent clearance [3]. Removal of circulating SAP with miridesap (see below) followed by treatment with complement fixing monoclonal antibody to SAP led to clinical improvement in human amyloidosis [8].

Over a half century ago, it was discovered that live or dead *C. albicans* injected into the thigh promoted systemic amyloidosis in mice [9,10,11]. Only recently has the connection between fungi, amyloid, and SAP been further investigated. SAP is a prominent component of fungal biofilms in deep seated infections [12,13]. We found a diminished host response to SAP-coated *Candida* yeasts and hyphae in 20 autopsy cases of disseminated candidiasis. This finding led us to hypothesize that SAP’s presence in deep seated fungal infections was deleterious to the host, likely dampening the host immune response [12,14,15]. We have shown that human SAP binds to functional amyloid on fungal cell surfaces and acts as an anti-opsonin [5]. Amyloid (without bound SAP) is naturally present on fungal cell surfaces and increases phagocytosis by host macrophages. The more amyloid expressed, the greater the phagocytosis [16]. Contrarily, SAP binding to a macrophage (or yeast cell) decreases phagocytosis and diminishes secretion of the inflammatory cytokines TNFα, IFγ, IL-6, and IL-17, and macrophages exposed to SAP secrete more of the anti-inflammatory cytokine, IL-10 [5]. Thus, in deep fungal infections SAP may function as an anti-opsonin and down regulator of the immune response [5].

There are few studies of the interactions of SAP with fungi other than our work [13]. Contrary to our findings it was reported that SAP opsonized *Sporothrix schenkii* [17]. Another study claimed human SAP enhanced “resistance” to *Aspergillus fumigatus* in mice [18]. However, the portal of entry of fungi in that study (via the meninges) does not mimic the pathogenesis of *Aspergillus* infection in the human. The improvement in mice noted by the investigators following SAP treatment was almost certainly due to the dampening effects that SAP has on cytokine secretion and attenuation of macrophage function rather than resistance to *Aspergillus* [18]. SAP-dependent amelioration of disease following the inhalation of *Aspergillus* conidia had been shown to involve a dampening of the host response [19]. *Aspergillus* conidia are heavily coated with functional amyloid [20] and bind SAP in vitro [13]. Pertinent to studies touting SAP as beneficial was a mouse model of COVID 19 pneumonia demonstrating that the associated cytokine storm was dampened by injecting human SAP [21]. In sum, SAP appears to dampen the immunological response to invasive fungi. That dampening of immunity may have either beneficial or malign effects for the host, depending on the specific conditions.

CPHPC (miridesap) is a bivalent small molecule drug that binds SAP at moderate affinity. However, its bivalent nature makes it highly avid. It successfully dissociates SAP from bound amyloids and induces depletion of serum SAP concentrations by turnover in the liver over a period of days to weeks [22]. Miridesap prevents binding of SAP to *C. albicans* and removes bound SAP from the fungi [16]. It has been suggested that miridesap treatment would hasten recovery from disseminated candidiasis, because it would increase phagocytosis and trigger a more robust immune response in a disease characterized by immune tolerance [23]. Therefore, we tested the effects of SAP and miridesap on disseminated candidiasis.

## 2. Materials and Methods

**Fungi.** *C. albicans* SC5413 was maintained on YPD agar (RPI, Mount Pleasant, IL, USA). Yeasts were prepared for intravenous injection by removing a loopful of fungi from the plate and adding it to 50 mL YPD liquid media (Life Technologies Corp., Carlsbad, CA, USA) in an Erlenmeyer flask. The flask was maintained at room temperature (26 °C) in a shaking water bath. After 24 h growth, 1 mL of YPD was placed in 50 mL fresh YPD and cultured for a further 24 h. Yeasts were washed by centrifugation in TRIS buffer with Ca^+2^ pH 7.8 (herein denoted as buffer). Yeast cells were counted, and dilutions performed to achieve the desired yeast cell concentration for intravenous injection. In some experiments washed yeasts were incubated with human SAP or human serum (30 μg/mL) for 1 h at 4 °C and allowed to come to room temperature before injection. To determine if miridesap (GlaxoSmithKline, Brentford, UK; a gift of M.B. Pepys) had fungicidal activity, *C. albicans* yeasts were cultured as above, washed in buffer and incubated at room temperature in buffer alone or buffer plus 5 mg/mL of miridesap for 1 h. Aliquots were then removed from each sample, plated on YPD agar and cultured for 48 h at room temperature. Colony counts were then determined for each sample.

**Mice.** Mice were housed in the University of Arizona Health Sciences animal facility. Water and food were available ad libitum. The animal use protocol for these experiments, #19-587, was reviewed and approved by the University of Arizona Institutional Animal Care and Use Committee. Balb/cJ males, 6–8 weeks old (Jackson Laboratory, Bar Harbor, ME, USA) were used (n = 260). Mice were housed in separate cages in groups of 5. All mice in a group received the same amount of yeast/gm weight. Mice were injected SQ with 2 mg pure human SAP or CRP (gifts of M.B. Pepys) or 100 µL buffer 18 h before day 0 when 4.5 × 10^4^ yeasts/gm of mouse were injected in the lateral tail vein (volumes of buffer varied from 90 to 130 µL). Mice continued to receive 1 mg of SAP or CRP SQ on days 1–4. Mice were checked 3 times/day and sacrificed when morbidly ill. In experiments using miridesap mice were divided into treatment groups and administered 3.0 × 10^4^ or 4.5 × 10^4^ yeast per gram of mouse on day 0. The miridesap treatment group received drinking water containing 5 mg/mL of miridesap beginning on day -10 and continued throughout the duration of the experiment [22].

**Histopathology.** Following sacrifice, the right kidney was removed and fixed in 10% formalin followed by 70% ethanol. Tissue was stained with hematoxylin and eosin and Gomori methenamine silver. Immunological stains included rabbit polyclonal antibody to human SAP (Biocare Medical, Pacheco, CA, USA); goat polyclonal antibody to human CRP, IHC-00613; and rabbit polyclonal to mouse SAP, PA5-81316; the immunogen is recombinant mouse serum amyloid P/APCS Protein (Met1-Asp244) (ThermoFisher, Waltham, MA, USA).

**Microscopy.** Microscopy was performed using the slide scanner, Leica Aperio AT2, (Leica Biosystems, Danvers, MA, USA) and photomicroscopy with the QuPath program (University of Edinburgh, https://qupath.github.io/ (accessed on 1 July 2022)).

**Statistics.** Data was analyzed using the 2 sample Mann–Whitney U test, paired Wilcoxon signed rank test, and Kaplan–Meier survival test (statskingdom.com). All experiments were repeated a minimum of 3 times with a minimum of 5 mice in each treatment group. Results were considered significant when *p* was <0.05. For most experiments, median time to death was 72–78 h for mice injected with yeasts only.

## 3. Results

**SAP-treated Mice.** We first tested the effect of SAP administered shortly before and after the intravenous injection of *C. albicans*. Human SAP (2 mg) was administered SQ 18 h before the injection of fungi thus providing serum levels of SAP approximating the human serum concentration of SAP [22]. Each day thereafter, mice received 1 mg SQ through day 4. The health of these mice was followed 3X daily and compared to mice that received a 100 µL SQ injection of buffer as well as yeast cells given intravenously. Mice given yeast cells only displayed scruffy coats and runny noses on day 1–2 after yeast cell injection, whereas mice administered SAP displayed normal coats and apparent good health until day 3 or 4 after yeast cell injection. No mice survived experiments when given 4.5 × 10^4^ yeasts/gram/mouse (see Figure 1 for Kaplan–Meier survival curve). The outcome was significantly different, *p* < 0.022 by Wilcoxon signed-rank test (n = 38 mice), with mice receiving yeast cells and SAP living longer than mice receiving *Candida* yeast cells alone.

Similar results were obtained when yeasts were soaked in human serum (which provides SAP) or SAP for 1 h, washed in buffer and then injected (denoted as SAP IV) or given intraperitoneally (IP). Mice receiving SAP survived longer than mice receiving yeast cells only, but no mice survived beyond a day or two over mice receiving yeast cells alone. Experiments with mice receiving yeast cells alone vs. yeast cells plus SAP given SQ, IV, or IP resulted in similar survival curves to those seen in Figure 1.

Thus, mice receiving SAP prior to intravenous yeast cell injection survived longer and did not die as soon as mice receiving yeast cells alone. No animals survived this dose of *Candida*, all were dead by 8–10 days.

**Histology of *Candida*-SAP-Mouse Kidney.**Figure 2 shows representative images of the evolution of the infection in the kidney, a targeted organ in mice and humans following fungal access to the vasculature [24]. Remarkably, after injection of yeasts intravenously, blood cultures of mice remain sterile for days, as previously observed [25,26]. However, in Figure 2B a budding yeast cell was detected within a collecting tubule in the kidney 30 min after intravenous injection and before attaching to tissue. (Thus, the yeast cell must have traversed the vascular system through the glomerulus and entered into the collecting system of the kidney). Figure 2C shows evolving abscesses in the kidney parenchyma 3 days after intravenous injection of yeast cells. Figure 2D,E show these abscesses consist of fungi stained with silver.

**CRP-treated Mice.** CRP and SAP are highly similar pentraxins, but generally have opposing activities [2,27]. CRP does not bind to yeast and its exogenous administration does not significantly affect macrophage phagocytosis of *C. albicans* [5]. Figure 3 demonstrates the rapid demise of mice administered yeast cells and CRP SQ compared to mice administered yeast cells and buffer SQ (*p <* 0.017). There were three separate experiments of mice administered yeast cells and CRP vs. mice administered yeast cells only and results were similar to those in Figure 3. Mice administered CRP became ill faster than mice not treated with CRP following the intravenous injection of *C. albicans* yeast cells. Their coats were scruffy and urine-stained on day 1 after injection, markedly different both from mice given yeast cells alone or mice treated with yeast cells and SAP SQ.

**Miridesap-treated Mice.** Addition of miridesap in the drinking water begun on day-10 had a significant effect upon survival of the mice. Miridesap-treated mice did not appear as sick as control mice following the intravenous injection of *C. albicans,* and they survived significantly longer than mice that received yeast cells only (*p* < 0.020). Figure 4 shows the results when 4 treatment groups of 5 mice were each injected intravenously with 4.5 × 10^4^ yeasts/gm/mouse at time 0. One group received miridesap in the drinking water beginning at day-10. Another group received miridesap and human SAP SQ; a third group received human SAP SQ; and the fourth group consisted of mice administered yeast cells alone. As can be seen, mice receiving miridesap survived the longest. Mice receiving yeast cells alone or yeast cells and SAP SQ did not survive as long.

Following these results, we pursued the effects of miridesap treatment alone. Figure 5 shows a typical survival plot of mice administered yeast cells alone vs. mice treated with miridesap and administered yeast cells. Results for mice injected with yeast at 4.5 × 10^4^ yeasts/g mouse (inset) were similar to those shown in Figure 4. However, when a smaller inoculum was used, the mice survived longer, and two of the miridesap-treated mice were still alive when the experiment was terminated. Miridesap had no fungicidal activity using a colony forming unit technique as explained in Section 2. Thus, treatment with miridesap significantly prolonged the life of mice and some were “cured” of candidiasis.

Figure 6 demonstrates that mouse SAP abundantly decorated fungal cell surfaces in kidney tissue.

## 4. Discussion

The pentraxin SAP is consistently bound to fungi in deep mycoses [13]. SAP binding to *C. albicans* is anti-opsonic, inhibiting macrophage phagocytosis and skewing macrophages to the anti-inflamatory M2 type [5]. However, the in vivo effects of SAP on the course of disseminated candidiasis have not been previously investigated.

We have assessed the effects of SAP and its binding inhibitor miridesap in a mouse model of disseminated candidiasis. We found that injecting human SAP at the time of injection of yeast cells into mice provided modest, short-lived benefit, whereas miridesap-mediated removal of SAP was more effective in ameliorating the infection. Neither SAP nor miridesap affected the growth of the fungus, so the effects on infection were directly attributable to host response to the SAP or its removal. The pro-inflammatory pentraxin CRP, on the other hand hastened morbidity and death, even though it does not bind to the fungi [27]. We propose that these experiments reflect primarily the immuno-modulatory activity of the pentraxins.

The IV injection of *C. albicans* at 4.5 × 10^4^ yeasts/gm/mouse led to infection of the kidney and colonization (Figure 2). Median survival time was about 75 h, and all mice died within 150–170 hrs. In mice also receiving SAP, there was a delayed onset of visible markers of disease, median survival time lengthened, and the survival time was extended by about 48 h (Figure 1). We attribute this improvement to SAP-mediated suppression of the inflammatory response. Previous results have shown that SAP binds to fungal cell surfaces, specifically to amyloid-like cross-β-bonded surface patches of adhesins. Such binding leads to repression of macrophage phagocytosis. In addition, macrophages exposed to SAP-coated *C. albicans* are skewed to the more tolerant M2 state, with greater secretion of anti-inflammatory cytokine IL-10 and decreased secretion of the pro-inflammatory cytokines, TNFα, IFγ, IL-6, and IL-17 [5]. SAP bound to the fungi dampened the immune response, but it allowed continued fungal colonization and did not abrogate long-term damage, which eventually killed the mice [28,29]. CRP, a homologous pro-inflammatory molecule mediated more rapid kidney colonization and mortality (Figure 2 and Figure 3).

Given that administration of SAP ameliorated symptoms of systemic candidiasis and prolonged life for 48 h, the results with the SAP binding inhibitor miridesap were paradoxical. Oral miridesap more than doubled the median survival time of infected mice to about three weeks (Figure 4 and inset in Figure 5). For mice infected at low dose (3.0 × 10^4^ yeasts/gm/mouse), miridesap, by removing SAP, facilitated resolution of the disease for some of the mice (Figure 5). Miridesap prevents SAP binding to yeast cell surface amyloids and dissociates previously bound SAP [2,16]. Miridesap-treated mice showed different outcomes than did mice not treated with miridesap in disseminated candidiasis. The difference must be due to effects of miridesap removing endogenous murine SAP, because these mice had no human SAP administered. SAP concentrations in mice vary widely, Balb/c mice have basal SAP concentrations of ~53 µg/mL [30], higher than either men or women. The SAP concentration can rise significantly following infection or inflammation and mice strains with lower basal concentrations of SAP undergo an increase in SAP to equal the concentrations of those strains with higher basal concentrations following an acute phase response [31]. This endogenous SAP would be anti-opsonic for *C. albicans* and skew the macrophages to the M2 state. Thus, miridesap-induced dissociation and turnover of SAP depleted this additional endogenous SAP, resulting in a more robust immune response that could resolve the infection. Figure 7 shows a model consistent with these results and with known effects of SAP on macrophages.

Therefore, the effects of injected SAP and miridesap imply that SAP has differing roles depending upon the time course in disseminated candidiasis. SAP binding to yeast cells and macrophages inhibits DC-SIGN-mediated phagocytosis, perhaps by masking the DC-SIGN ligands on the cell surface [32,33]. At the same time, SAP binding to FcγR1 receptors leads to skewing towards the non-inflammatory M2 state for macrophages [16,34]. We posit that SAP’s anti-inflammatory activity is initially beneficial to the host but allows continued proliferation of the fungi. As the disease progresses, SAP becomes deleterious to the immune response that would attack the fungus. Miridesap treatment allows the initial dampening of the response, but eventually depletes SAP enough to promote immune response leading to more positive outcomes for the host, as illustrated in Figure 7.

## 5. Conclusions

Disseminated candidiasis in humans has a high morbidity and mortality rate even with current antifungal therapies and the number of cases is increasing worldwide. New modalities of therapy may be helpful and such an approach is the use of miridesap to bind and remove SAP that dampens the immune response. SAP is an invariable, major component of pathologic amyloid deposits, either intra- or extracellularly. A striking feature of amyloidotic deposits is that the host does not mount an immune response to the pathologic amyloid, due to bound SAP. SAP binds avidly to the functional amyloid on pathologic fungi and is removed by miridesap. Miridesap may have potential as adjuvant therapy of deep-seated fungal infections.

## Figures and Tables

**Figure 1 pathogens-11-01304-f001:**
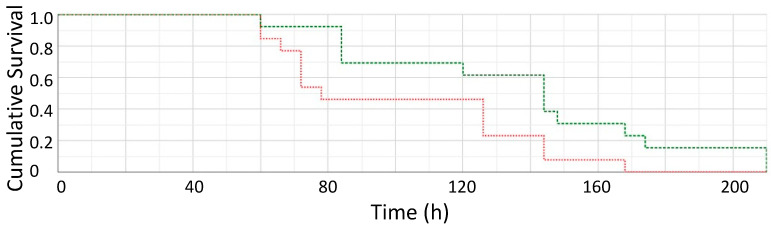
Kaplan–Meier Survival plot (in hours). Mice receiving yeast cells alone (red) vs. mice receiving yeast cells and SAP SQ (green); all mice received 4.5 × 10^4^ yeasts/gram/mouse intravenously on day 0. (n = 38 mice).

**Figure 2 pathogens-11-01304-f002:**
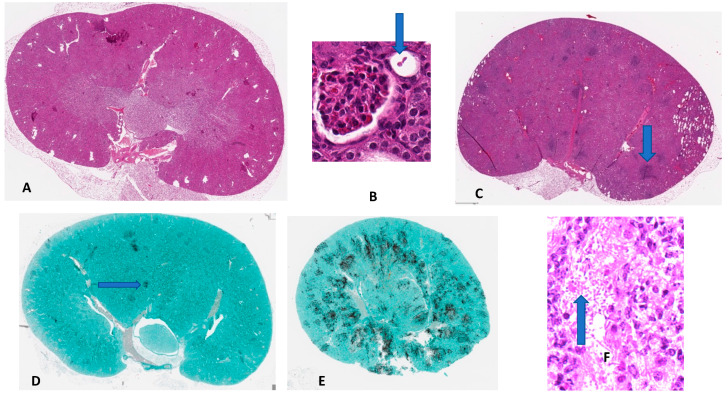
Histopathology of murine disseminated candidiasis. (**A**) Uninfected right kidney; H&E (×1.07 magnification); (**B**) Kidney 30 min after intravenous injection; a budding yeast (arrow) is seen within a collecting tubule having penetrated the glomerulus after traversing the vasculature following injection; (H&E × 41). (**C**) Right kidney, 3 days after receiving *C. albicans* only; dark condensation of tissue (arrow) with round cells and polymorphonuclear leukocytes are scattered throughout the parenchyma where later, abscesses will form; (H&E × 1.07). (**D**) Kidney 3 days after receiving *C. albicans* intravenously; the dark staining areas (arrow) are fungi in the early stages of an abscess; (GMS × 1.07). (**E**) Kidney 3 days after injection of *C. albicans* along with CRP; more fungi are apparent as well as many more abscesses than in (**D**); (GMS × 1.07). (**F**) Kidney tissue of mouse in (**E**) stained with H&E showing polymorphonuclear leukocytes and round cells surrounding yeast cells and hyphae (arrow); ×41.

**Figure 3 pathogens-11-01304-f003:**
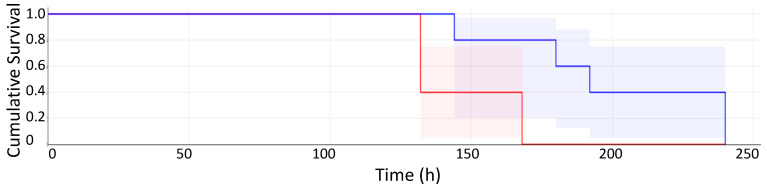
Kaplan–Meier Survival plot (in hours with confidence intervals) of mice administered yeast cells only (blue) vs. mice administered yeast cells and CRP SQ (red). Typical survival curve; log-rank test shows survival to be significantly different, *p* < 0.017. Five mice in each group; 4.5 × 10^4^ yeasts/g/mouse given intravenously at day 0. Although this experiment showed atypically long survival of the mice, all repetitions gave accelerated death for mice injected with CRP.

**Figure 4 pathogens-11-01304-f004:**
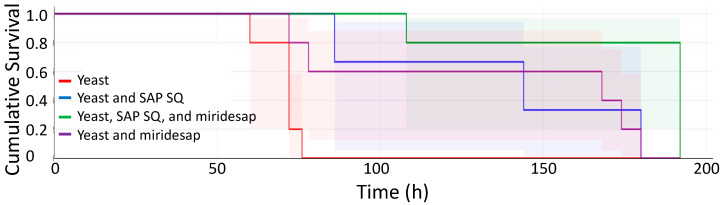
Kaplan–Meier Survival plot (in hours) with confidence limits of 4 treatment groups injected with *C. albicans*. Each group of 5 mice was injected with 4.5 × 10^4^ yeasts/gram/mouse intravenously at day 0: mice administered yeast cells (red); mice administered yeast cells and SAP SQ (blue); mice administered yeast cells and SAP SQ and given miridesap in the drinking water (green); and mice administered yeast cells and miridesap in the drinking water but no SAP (purple); 4.5 × 10^4^ yeasts/gm of mouse were administered intravenously at day 0.

**Figure 5 pathogens-11-01304-f005:**
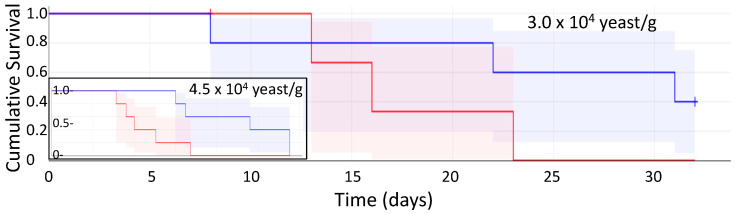
Kaplan–Meier Survival plot (in days) with confidence limits of mice administered yeast cells alone (red) vs. mice administered yeast cells and receiving miridesap in drinking water (blue); five mice in each group; 3.0 × 10^4^ yeasts/gm of mouse intravenously at day 0. Mice were weighed daily. Two (2) mice in the miridesap group recovered from disseminated candidiasis and were regaining weight when the experiment was terminated. The two groups were significantly different in survival time. **Inset:** Survival plot (in days) with confidence limits of mice administered yeast cells alone (red; all mice died by day 7) and mice administered yeast cells and receiving miridesap in drinking water (blue; all mice died by day 12); five mice in each group; 4.5 × 10^4^ yeasts/gm of mouse intravenously at day 0. The time scale on the X-axis is the same for both main plot and the inset. The survival time with confidence limits is shown and is significantly different for each group, *p* < 0.020.

**Figure 6 pathogens-11-01304-f006:**
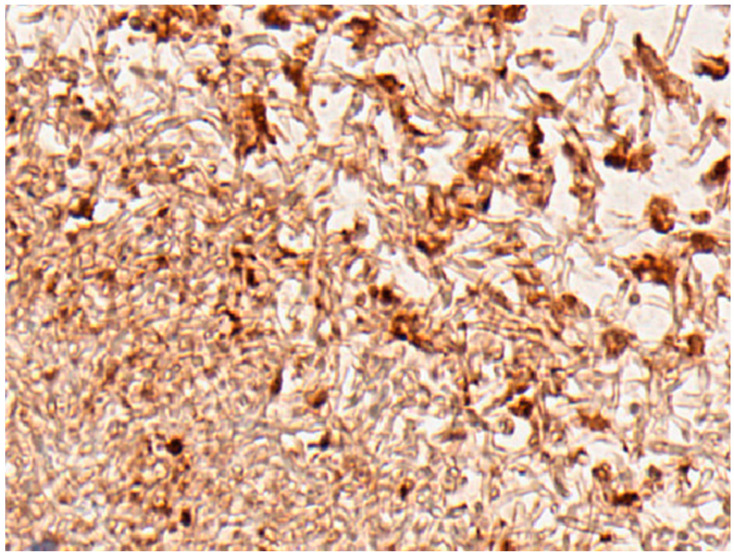
Abscess in a mouse 4 days after intravenous *Candida albicans* injection with 4.5 × 10^4^ yeasts/gm/mouse. Hyphae and yeast cell surfaces are decorated with mouse SAP (brown). This tissue section was chosen because no mouse tissue is seen in the abscess. Mouse tissue would show extensive SAP since it is an integral component of basement membrane.

**Figure 7 pathogens-11-01304-f007:**
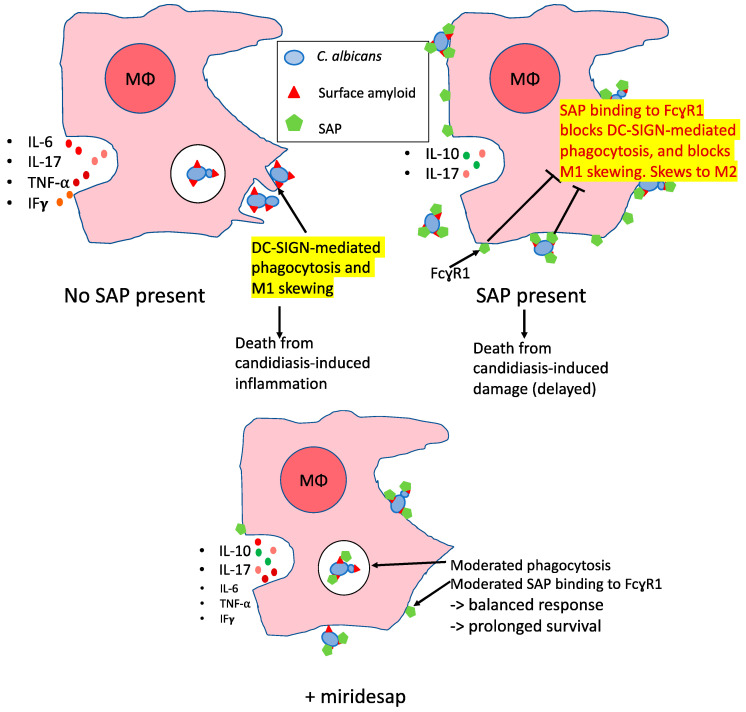
Speculative model of effects of SAP and miridesap on macrophage response to candidiasis. Upper Left: *Candida* amyloid PAMPs are phagocytosed through macrophage DC-SIGN, leading to M1 skewing and secretion of pro-inflammatory cytokines. Upper Right: SAP mediates masking of *Candida* amyloid PAMPs and induces M2 skewing as a result of SAP binding to macrophage FcγR1, leading to secretion of IL-10 [5,32,33]. Lower: Miridesap moderates the effects of SAP and allows a more balanced response including moderate pro- and anti-inflammatory cytokine secretion [16,34].

## Data Availability

Not applicable.

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
