# Peer review of "The Paradoxical Effects of Serum Amyloid-P Component on Disseminated Candidiasis"

_pathogens, 2022, doi:10.3390/pathogens11111304_

Round 1

Reviewer 1 Report

This is an interesting manuscript evaluating the effect of serum amyloid-P component (SAP) and miridesap administration on Candida albicans infections in mice. The manuscript contains some interesting data, provides novel insights into working mechanism of these compounds, and the paper is fairly well written.

Some comments/suggestions for improvement:

1. L71. Sporothrix

2. L98. Why is C. albicans grown at 26°C?

3. Presentation of data in Figures with Kaplan-Meier plots. The experiments should be described in a more consistent way. For instance, only in the Fig.1 experiment the total number of mice (N=38) is mentioned in the text and the use of Wilcoxon (not Wilcoxin!) rank test is mentioned only in this legend but there are no confidence intervals as given in the other figures.

4. There are rather large differences between the results in the different figures, which points to problems in experimental reproducibility, see for instance the differences between the control groups in Figs 1 & 3, and the control group and miridesap treatment in Fig.4 vs Fig. 5 (inset). A proper explanation for these differences should be provided.

5. Fig.5. Not clear if the data are explained in the text and figure legend is missing or, reversely, that the data is only explained in the legend but not in the text.

6. L.214. Figure 6 is not a survival plot.

7. L.264-266. Same sentence is given two times.

8. L.273-274. "Because miridesap-treated mice showed different outcomes of candidiasis than mice without added SAP". Not clear what is meant. Are the authors comparing miridesap with SAP to miridesap without added SAP, or perhaps miridesap without added SAP to the control group?? Specify.

9. Why is a concentration of 5 mg/ml miridesap chosen. Would the obtained results perhaps prompt experiments with different doses of this compound?

Author Response

Response to Reviewer 1

General Response: We thank the Reviewer for the corrections and comments. We have changed the title of the paper because it was somewhat confusing. We have edited and changed the Figure legends to make them more consistent.

  1. L71. Sporothrix

Response: corrected spelling

  1. L98. Why is C. albicans grown at 26°C?

Response: This is to avoid germ tube and hyphal morphology; the fungus is uniformly in yeast cell morphology at this temperature in YPD,

  1. Presentation of data in Figures with Kaplan-Meier plots. The experiments should be described in a more consistent way. For instance, only in the Fig.1 experiment the total number of mice (N=38) is mentioned in the text and the use of Wilcoxon (not Wilcoxin!) rank test is mentioned only in this legend but there are no confidence intervals as given in the other figures.

Response: The figure legends are now presented in a consistent way. We changed the terminology to reflect that mice received yeast cells alone or mice received yeast cells along with SAP, CRP or miridesap. We have deleted the word “control” throughout, including within Figure 4.

  1. There are rather large differences between the results in the different figures, which points to problems in experimental reproducibility, see for instance the differences between the control groups in Figs 1 & 3, and the control group and miridesap treatment in Fig.4 vs Fig. 5 (inset). A proper explanation for these differences should be provided.

Response: For mice injected with yeast alone at 4.5 x 104 yeasts/g mouse, the median time to death ranged from 72-78 hours, with the exception of Fig. 3. We have added the requested language in lines 134-135 and 200-201.  We have also labeled Fig. 4 with the inoculum sizes to clarify the difference in survival at lower dose.

  1. Fig.5. Not clear if the data are explained in the text and figure legend is missing or, reversely, that the data is only explained in the legend but not in the text.

Response: We have added a summary of the results to the text (Lines 222-227)

  1. L.214. Figure 6 is not a survival plot.

Response: Correction made.

  1. L.264-266. Same sentence is given two times.

Response: Repetition was deleted.

  1. L.273-274. "Because miridesap-treated mice showed different outcomes of candidiasis than mice without added SAP". Not clear what is meant. Are the authors comparing miridesap with SAP to miridesap without added SAP, or perhaps miridesap without added SAP to the control group?? Specify.

Response: This is reworded as follows: Miridesap-treated mice showed different outcomes than did mice not treated with miridesap in disseminated candidiasis. The difference must due to effects of miridesap removing endogenous murine SAP, because these mice had no human SAP administered.” (now lines 290-293)

  1. Why is a concentration of 5 mg/ml miridesap chosen. Would the obtained results perhaps prompt experiments with different doses of this compound?

Response: This procedure is now referenced to [22], (line 122) where dosing at this concentration is recommended.

Reviewer 2 Report

Yin and Yang: The Paradoxical Ameliorative Effects of Serum Amyloid-P Component and its Inhibitor, Miridesap, on Disseminated Candidiasis

The article describes the role of serum amyloid P component (SAP) in disseminated candidiasis and the role of the SAP binding inhibitor, miridesap, in modulation of disease outcomes. Overall, the manuscript describes potentially novel and interesting findings. However, further experimental work is required before it can be considered for publication. For example, there is a lack of biological investigations into the mechanistics of the proposed hypotheses and instead a serious of KM plots is reported. In addition, the experimental design, with respect to performing and/or reporting of appropriate controls, requires attention and would help the reader determine the true biological significance of the reported data. More detailed information can be found below in the ‘major’ comments section.

Major

Figure 1

Legend lacking in detail which makes interpretation of experimental design difficult. It would be more beneficial and easier for the reader to show all groups on the KM plot. I’m assuming this included (i) sham, (2) SAP alone (no C. albicans), (3) C. albicans only and (4) C. albicans + SAP. 2 mg SAP was provided 18 h before and the authors state ‘this provided serum levels approximating the human serum concentration’ yet there is no data to support this. The authors state that ‘experiments with control mice v SAP given by different routes (SQ, IV, IP) gave results similar to Figure 1’. The experimental design for these experiments was very different (yeast soaked in SAP) and once again it would benefit as being presented as a standalone panel/figure. It is also hard to determine what the control groups were in these set of experiments. In the text it says that n=38 but the legend sates that there were 18 matched pairs. In addition, as multiple experiments were performed how do the n numbers equate to these experiments? It is hard therefore to determine statistical significance of the findings.

Figure 2

The authors state that shortly after infection of yeasts IV blood cultures remain sterile for days, yet they report a budding yeast in the collecting tubule after 30 min. What is the rationale for this? The figures shown are on the small side and hard to view. I would suggest magnified images are shown as well to see more structured features, is there evidence of candida hyphae? Images reported seem to be ‘one offs’ are these representative images of multiple mice in each group? Where are the appropriate controls, for example does administration of SAP alone affect pathology of the kidneys?

Figure 3

The comments made for figure 1 are also applicable here. There is a lack of detail and also lack of data for appropriate control groups. Once again, I suggest reporting all data for at least the following (i) sham, (2) CRP alone (no C. albicans), (3) C. albicans only and (4) C. albicans + CRP. In addition, if multiple repeats of the experiment are performed then report all results combined, not just a ‘representative’ of one repeat as it seems to suggest here (Lines 186-187).

Figure 4

What is the rationale for administration of miridesap 10 days before infection and in the drinking water? What (if any) was the long-term effects of miridesap alone administered in the water (absence of infection)

Figure 5

Once again appropriate controls are not shown. Mice given C. albicans are not ‘controls’. The figure does not show the effect of miridesap treatment ‘alone’

Figure 6

The staining looks non-specific. No controls are shown so hard to determine

Figure 7

Although interesting hypothesise are proposed, the data presented in the manuscript do not support this and further experimental work is required.

Author Response

Response to Reviewer 2

General Response: The Reviewer mentioned that more work needed to be done on understanding the mechanisms of action of SAP and devising appropriate experiments. Over the years we have carefully worked out the relationship of human SAP and fungi, in particular, its binding to Candida albicans. In a detailed prior study, (reference 5), we delineated the interaction of SAP and human macrophages and SAP’s effect on human macrophages and cytokine production and the phagocytosis of C. albicans. In another recently published paper, we looked at the effect that miridesap had on SAP bound to fungi (reference 16). The study we report here was an attempt to test our hypotheses in vivo that SAP downregulated the host immune response to fungi coated with SAP. We found that at the beginning of murine disseminated candidiasis SAP provided some benefit by probably dampening the cytokine storm whereas, removing SAP by use of miridesap, mice lived longer and in some cases seemed to survive what would have been a lethal dose of Candida.

Figure 1

Legend lacking in detail which makes interpretation of experimental design difficult. It would be more beneficial and easier for the reader to show all groups on the KM plot. I’m assuming this included (i) sham, (2) SAP alone (no C. albicans), (3) C. albicans only and (4) C. albicans + SAP. 2 mg SAP was provided 18 h before and the authors state ‘this provided serum levels approximating the human serum concentration’ yet there is no data to support this. The authors state that ‘experiments with control mice v SAP given by different routes (SQ, IV, IP) gave results similar to Figure 1’. The experimental design for these experiments was very different (yeast soaked in SAP) and once again it would benefit as being presented as a standalone panel/figure. It is also hard to determine what the control groups were in these set of experiments. In the text it says that n=38 but the legend sates that there were 18 matched pairs. In addition, as multiple experiments were performed how do the n numbers equate to these experiments? It is hard therefore to determine statistical significance of the findings.

Response: We agree that the terminology we used, for instance the word “control”, was not appropriate and confused both reviewers. The figure texts are now presented in a consistent way. We changed the terminology to reflect that mice received yeast cells alone or mice received yeast cells along with SAP, CRP or miridesap. We deleted the word “control”. Figure 1 is now simply a survival plot for visualization of the 38 animals as a group.

Figure 2

The authors state that shortly after infection of yeasts IV blood cultures remain sterile for days, yet they report a budding yeast in the collecting tubule after 30 min. What is the rationale for this? The figures shown are on the small side and hard to view. I would suggest magnified images are shown as well to see more structured features, is there evidence of candida hyphae? Images reported seem to be ‘one offs’ are these representative images of multiple mice in each group? Where are the appropriate controls, for example does administration of SAP alone affect pathology of the kidneys? 

Response: The images are representative, and we have now noted this. We have also added clarifying language in lines 165-168 for Fig 2b. We also added references to the support the rapid clearance of yeast from the bloodstream. The SAP alone control would be equivalent to the Fig. 2A, due to endogenous mouse SAP.

Figure 3

The comments made for figure 1 are also applicable here. There is a lack of detail and also lack of data for appropriate control groups. Once again, I suggest reporting all data for at least the following (i) sham, (2) CRP alone (no C. albicans), (3) C. albicans only and (4) C. albicans + CRP. In addition, if multiple repeats of the experiment are performed then report all results combined, not just a ‘representative’ of one repeat as it seems to suggest here (Lines 186-187).

Response: We hope that the changes in language to eliminate the word “control” throughout will satisfy theis point (please see the response to the critique of Fig. 1). Sham-infected mice will live for 1-3 years, and are not a comparable population. Injecting SAP or CRP did not shorten the live span or affect the mice in any measurable way unless they were injected with Candida yeast cells.

Figure 4

What is the rationale for administration of miridesap 10 days before infection and in the drinking water? What (if any) was the long-term effects of miridesap alone administered in the water (absence of infection)

Response: Please see point 9 under reviewer 1. This protocol has now been referenced. As mentioned in lines 84-91, miridesap depletes the mice of circulating SAP and any SAP attached to the fungi, thus allowing the host response to attack the fungi. The drug when administered in drinking water takes several days to achieve the maximum effect. SAP when bound to various abnormal or foreign proteins or tissue causes the host to treat the SAP-coated molecules as immunologically inert. Thus, by using miridesap we are removing SAP that prevents an immune response. Reference 22, which describes miridesap explains this in great detail. Miridesap is harmless to mice and humans and humans have been administered the drug for years without harm. Apparently, humans and mice can live without SAP, but it is interesting that SAP is found throughout phylogeny and no human has been described without SAP and there have been no major functional mutations reported in human SAP.

Figure 5

Once again appropriate controls are not shown. Mice given C. albicans are not ‘controls’. The figure does not show the effect of miridesap treatment ‘alone’ 

Response: We agree with the Reviewer and have changed the nomenclature and explanations. Please see our response to Reviewer’s Figure 1 (above).

Figure 6

The staining looks non-specific. No controls are shown so hard to determine

Response: We know from our prior studies of human SAP and Candida and use of flow cytometry that the staining is quite specific. SAP is binding to functional amyloid on the fungal surface (see references 5,12 and 13). This photograph shows hyphal structures infecting the mouse kidney that are coated with murine SAP (brown tinge). The SAP is abundantly attached to the surface of the fungus. This tissue section was chosen for the reason that it shows only fungi in an abscess and no host tissue is complicating the interpretation. Serum amyloid P component would be seen extensively in the kidney where is is found in the basement membrane. The polyclonal antibody had a mistake in its catalog number and this had been corrected as well as the target protein, as explained in Methods.

Figure 7

Although interesting hypothesise are proposed, the data presented in the manuscript do not support this and further experimental work is required.

Response: As mentioned in the text, the figure is a testable model consistent with results herein and also with published results of SAP effects on pathogen and innate immunity. Additional mechanistic experiments are beyond the scope of the current paper, and unfortunately are not within our financial capability. We believe the experiments to be supportive of the model, and therefore, we feel that retaining the figure will both explain our paradoxical results and stimulate further experimentation by others.